# The Semi-Hyperbolic Distribution and Its Applications

**Roman V. Ivanov** 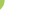

Laboratory of Control under Incomplete Information, V.A. Trapeznikov Institute of Control Sciences of RAS, Profsoyuznaya 65, 117997 Moscow, Russia; roivanov@yahoo.com

**Abstract:** This paper studies a subclass of the class of generalized hyperbolic distribution called the semi-hyperbolic distribution. We obtain analytical expressions for the cumulative distribution function and, specifically, their first and second lower partial moments. Using the received formulas, we compute the value at risk, the expected shortfall, and the semivariance in the semi-hyperbolic model of the financial market. The formulas depend on the values of generalized hypergeometric functions and modified Bessel functions of the second kind. The research illustrates the possibility of analysis of generalized hyperbolic models using the same methodology as is employed for the well-established variance-gamma model.

**Keywords:** generalized hyperbolic distribution; semi-hyperbolic distribution; hypergeometric function; Bessel function; value-at-risk; expected shortfall; semivariance





## 1. Introduction

In this paper, we explore a subclass of the generalized hyperbolic (GH) distribution, which we term the semi-hyperbolic distribution. The class of GH distribution was primarily introduced by Barndorff-Nielsen [1] for the investigation of the physics of dune movements. This distribution is infinitely divisible, and therefore, it originates from corresponding Lévy processes. The widely discussed subclasses of the GH distribution are the normal–inverse Gaussian (NIG) and the hyperbolic distribution. We call the considered subclass of distributions semi-hyperbolic because their probability density functions are close to the densities of the hyperbolic distribution. Presently, an important area of the applications of GH distribution is financial index modeling; see the work by Eberlein [2]. Next, we introduce the family of GH distribution in more detail. Specifically, we start from the discussion of the two most prominent members of the class of GH distribution: the NIG and hyperbolic distributions. We consider below several research papers in which the theoretical properties of these distributions are investigated, and the financial data are calibrated in the related market models.

The NIG distribution was first discussed in terms of financial applications by Barndorff-Nielsen [3] and Rydberg [4]. The properties of the NIG distribution are summarized in Barndorff-Nielsen [5]. Different approximations of the NIG Lévy processes are provided by Pacheco-González [6] and by Chapter 3 of Rasmus [7] and Benth et al. [8]. Computational methods in the NIG model for various problems of mathematical finance have been developed by Aguilar [9], Ivanov and Temnov [10], and Venter and de Jongh [11]. It has been shown by Mabitsela et al. [12] that the NIG distribution fits well with the equity behavior at the Johannesburg Stock Exchange.

The hyperbolic distribution was examined as a pattern for daily German stock index returns, primarily by Eberlein and Keller [13]. Eberlein et al. [14] confirmed the use of the hyperbolic model for the simulation of the NYSE composite. Küchler et al. [15] presented an empirical study that affirmed that the hyperbolic distribution fitted well to the stock returns of German companies listed on the DAX and the FAZ indices. Bauer [16] showed that the symmetric hyperbolic distribution estimates the 1% value-at-risk substantially better

than the normal distribution. Dorić and Nikolić-Dorić [17] investigated how accurately the hyperbolic distribution described BELEX15 index returns.

Furthermore, we consider the two distributions that are very close to GH ones. They are the variance-gamma and skewed Student's t distributions. As is revealed in Eberlein and von Hammerstein [18], the gamma and the inverse gamma distributions can be understood as limiting cases of the generalized inverse Gaussian (GIG) distribution. Therefore, since the GIG probability densities are the mixing densities for the GH distribution, the variance-gamma distribution and the skewed Student's t distribution can be referred to as the limiting cases of the GH ones. The variance-gamma and skewed Student's t distributions are very popular in financial applications. Madan and Seneta [19] modeled the indices of the Sydney Stock Exchange using the symmetric variance-gamma distribution and introduced a compound Poisson approximation for this distribution. Daal and Madan [20] showed that the variance-gamma model fitted well for the exchange rate dynamics of German financial markets. Rathgeber et al. [21] evaluated the parameters of the variance-gamma distribution gauging Dow Jones index returns. Employing the variance-gamma model, Wallmeier and Diethelm [22] calibrated the Swiss market for structured financial products. Alvarez and Baixauli [23] showed that the Student's t model for index returns provides better results in value-at-risk estimation than the normal, the logistic, and the Edgeworth–Sargan models for several market variables. Using maximum likelihood estimation, Aas and Haff [24] calibrated the parameters of the skewed Student's t distribution for international bonds and Norwegian indices. The theoretical properties of the skewed Student's t distribution were investigated by Finlay and Seneta [25]. Müller and Righi [26] confirmed that the skewed Student's t model has good performance for value-at-risk prediction based on the Ibovespa index statistics.

Analytical results for distribution functions, risk measures, and option prices are obtained in the mentioned models using generalized hypergeometric functions, which are computed very fast. For their properties and connections with other special functions, see Rathie and de Sena Monteiro Ozelim [27], Choi et al. [28], and Srivastava [29]. We refer to Madan et al. [30], Ano and Ivanov [31], and Ivanov [32] regarding the variance-gamma model, to Ivanov [33] regarding the skew Student's t model, and to Ano and Ivanov [31] and Ivanov and Temnov [10] regarding the NIG one. At the same time, modern research papers that analyze data from financial markets often recommend the simulation of index returns with GH distribution as well as the above-mentioned ones.

The following works study the class of GH distribution without the suggestion that one of the parameters of the distribution is fixed. In particular, all parameters of the GH distribution are estimated in those papers about financial market data. Daskalaki and Katris [34] reported that the GH distribution is the most successful choice for European stock market modeling. Baciu [35] showed that the GH model is the best fit for Bucharest Stock Exchange returns. Rathgeber et al. [36] extended the simplified method of moments for the problem of the estimation of the parameters of the GH process. Balter and McNeil [37] calibrated symmetric GH distribution based on S&P500 index statistics. An extension of the family of GH distributions was introduced in Klebanov and Rachev [38]. Mixtures of the GH distribution have been studied by Han and Yin [39].

Semi-hyperbolic distribution is a subclass of GH distribution, which is determined by the parameter $\lambda = 0$ for this subclass (see Section 2 in this paper for details). In particular, as is shown in Table 12 of Daskalaki and Katris [34], this value of $\lambda$ conforms to the movements of the OMXH25 and SAX indices. The aim of this paper is to obtain analytical results for the theoretical characteristics of the semi-hyperbolic distribution and to apply them to the computation of the basic monetary risk measures in the semi-hyperbolic model.

The rest of this paper is organized as follows. Section 2 describes the class of GIG distribution, including the semi-hyperbolic inverse Gaussian (SHIG) one, which is the mixing distribution for the semi-hyperbolic distribution. In Section 3, we present the probability density function and the moments of the semi-hyperbolic distribution, and formulate the main theoretical results of the work. The values of the monetary risk measures

are given in Section 4. Examples of the calculation of the obtained formulas are presented in Section 5.

## 2. Materials and Methods

As mentioned in the previous section, the semi-hyperbolic distribution is a normal mean-variance mixture, where the mixing density is the semi-hyperbolic inverse Gaussian (SHIG) distribution. The SHIG distribution is a member of the class of generalized inverse Gaussian (GIG) distributions. This class introduced in Good [40], Sichel [41] and Barndorff-Nielsen [1] for the aim of statistical modeling in variety of scientific areas. Properties of the family of GIG distributions are summarized in Eberlein and von Hammerstein [18]. Methods of the parameter estimation for these distributions were discussed by Tsai et al. [42] and Lee and Whitmore [43].

A generalized inverse Gaussian distribution $\mathrm{GIG} = \mathrm{GIG}(\lambda, \delta, \alpha, \beta)$ has the probability density function

$$f_{\mathrm{GIG}(\lambda,\delta,\alpha,\beta)}(x) = \frac{\left(\alpha^2 - \beta^2\right)^{\frac{\lambda}{2}} \delta^{-\lambda}}{2\mathrm{K}_\lambda\left(\delta\sqrt{\alpha^2 - \beta^2}\right)} x^{\lambda-1} e^{-\frac{\delta^2}{2x} - \frac{(\alpha^2 - \beta^2)x}{2}}, \tag{1}$$

where

$$\mathrm{K}_{\varsigma_1}(\varsigma_2) \tag{2}$$

is the modified Bessel function of the second kind (on its properties, see Chapters 9 and 10 of Abramowitz and Stegun [44]). The restrictions on the parameters in (1) are $\delta > 0$ and $|\beta| < \alpha$.

Next, we consider important examples of the GIG distributions and their limiting cases (see Eberlein and von Hammerstein [18] for details), which are mentioned in the Introduction.

**Example 1.** *Inverse Gaussian distribution. Since*

$$\mathrm{K}_{\pm\frac{1}{2}}(\varsigma) = \sqrt{\frac{\pi}{2\varsigma}} e^{-\varsigma} \tag{3}$$

*with respect to formula 8.469.3 of Gradshteyn and Ryzhik [45], we have in the case of $\lambda = -\frac{1}{2}$ that*

$$f_{\mathrm{GIG}\left(-\frac{1}{2},\delta,\alpha,\beta\right)}(x) = \frac{\delta}{\sqrt{2\pi x^3}} e^{-\frac{\left(x\sqrt{\alpha^2-\beta^2}-\delta\right)^2}{2x}} = f_{\mathrm{IG}\left(\sqrt{\alpha^2-\beta^2},\delta\right)}(x),$$

*where IG has the inverse Gaussian distribution, that is*

$$\mathrm{IG} = \inf\left\{t \geq 0: \ W_t + t\sqrt{\alpha^2 - \beta^2} = \delta\right\}$$

*for a Wiener process $(W_t)_{t\geq 0}$ (see Subsection 2.2.1 of Rasmus [7] or Chapter III.1d.3 of Shiryaev [46]).*

**Example 2.** *Hyperbolic inverse Gaussian distribution. If $\lambda = 1$, then*

$$f_{\mathrm{GIG}(1,\delta,\alpha,\beta)}(x) = \frac{\left(\alpha^2 - \beta^2\right)^{\frac{1}{2}}}{2\delta\mathrm{K}_1\left(\delta\sqrt{\alpha^2 - \beta^2}\right)} e^{-\frac{\delta^2}{2x} - \frac{(\alpha^2 - \beta^2)x}{2}} = f_{\mathrm{HIG}(\alpha^2 - \beta^2, \delta^2)}(x),$$

*where HIG has the hyperbolic inverse Gaussian distribution with parameters $\alpha^2 - \beta^2$ and $\delta^2$, see (10) in Eberlein and Keller [13].*

**Example 3.** *Gamma distribution. The modified Bessel function of the second kind has the asymptotics (see 9.6.9 of Abramowitz and Stegun [44]) for $\varsigma_1 \neq 0$*

$$K_{\varsigma_1}(\varsigma_2) \sim \frac{2^{|\varsigma_1|-1}\Gamma(|\varsigma_1|)}{\varsigma_2^{|\varsigma_1|}} \qquad as \; \varsigma_2 \to 0, \tag{4}$$

*where $\Gamma(\varsigma)$ is the gamma function. It follows immediately from (4) that if $\lambda > 0$, then*

$$\frac{\eta^\lambda \delta^{-\lambda}}{2K_\lambda(\delta\eta)} \to \frac{\eta^{2\lambda}}{2^\lambda\Gamma(\lambda)} \qquad as \; \delta \to 0. \tag{5}$$

*Therefore, we have from (1) and (5) that*

$$\lim_{\delta \to 0} f_{\text{GIG}(\lambda,\delta,\alpha,\beta)}(x) = \frac{(\alpha^2-\beta^2)^\lambda}{2^\lambda\Gamma(\lambda)} x^{\lambda-1} e^{-\frac{(\alpha^2-\beta^2)x}{2}} = f_{Y\left(\lambda,\frac{\alpha^2-\beta^2}{2}\right)}(x),$$

*where Y has the gamma distribution with the parameters $\lambda$ and $\frac{\alpha^2-\beta^2}{2}$.*

**Example 4.** *Inverse gamma distribution. When $\lambda < 0$, it follows from (4) that*

$$\frac{\eta^\lambda \delta^{-\lambda}}{2K_\lambda(\delta\eta)} \to \frac{2^\lambda}{\delta^{2\lambda}\Gamma(-\lambda)} \qquad as \; \eta \to 0 \tag{6}$$

*and we obtain from (1) and (6) that*

$$\lim_{|\beta| \to \alpha} f_{\text{GIG}(\lambda,\delta,\alpha,\beta)}(x) = \frac{\delta^{2|\lambda|}}{2^{|\lambda|}\Gamma(|\lambda|)} x^{\lambda-1} e^{-\frac{\delta^2}{2x}} = f_{\text{IY}\left(|\lambda|,\frac{\delta^2}{2}\right)}(x),$$

*where IY has the inverse gamma distribution with the parameters $|\lambda|$ and $\frac{\delta^2}{2}$.*

In this paper, we consider the case of $\lambda = 0$ in (1). The semi-hyperbolic inverse Gaussian distribution SHIG $=$ SHIG($\alpha, \beta, \delta$) has the three parameters $0 \leq |\beta| < \alpha$, $\delta > 0$ and the probability density function

$$f_{\text{SHIG}(\alpha,\beta,\delta)}(x) = f_{\text{GIG}(0,\delta,\alpha,\beta)}(x) = \tag{7}$$
$$= \frac{1}{2K_0\left(\delta\sqrt{\alpha^2-\beta^2}\right)} x^{-1} e^{-\frac{\delta^2}{2x}-\frac{(\alpha^2-\beta^2)x}{2}}.$$

The semi-hyperbolic distribution SH $=$ SH($\mu, \sigma, \alpha, \beta, \delta$) is defined as the normal mean-variance mixture (see Barndorff-Nielsen [1], Section 3.2 of McNeil et al. [47] and Chapter III.1d of Shiryaev [46] on the theory of mean-variance mixtures) with the SHIG mixing density, that is

$$SH = \mu + \beta SHIG + \sigma N \sqrt{SHIG}, \tag{8}$$

where $\mu \in \mathbb{R}$, $\sigma > 0$ and N $=$ N$(0,1)$ is the standard normal random variable independent with the SHIG one. Further, we study the SH distribution based on the definition (8).

## 3. Preliminary Formulas

In this section, we specify formulas that have been first received in general forms by Barndorff-Nielsen and Stelzer [48], Scott et al. [49] and given in Chapters 9.4 and 9.5 of Paolella [50] for the GIG and GH distributions. These identities are also required since the GH distribution was defined in those works with $\sigma = 1$ in the mixtures of type (8).

Let

$$C = \frac{1}{2K_0\left(\delta\sqrt{\alpha^2 - \beta^2}\right)}. \tag{9}$$

The characteristic function $\varphi_{\text{SHIG}(\alpha,\beta,\delta)}(u)$ of the SHIG distribution is computed as

$$\varphi_{\text{SHIG}(\alpha,\beta,\delta)}(u) = \int_0^\infty e^{iux} f_{\text{SHIG}(\alpha,\beta,\delta)}(x) = C\int_0^\infty x^{-1} e^{-\frac{\delta^2}{2x} - \frac{(\alpha^2 - \beta^2 - 2iu)x}{2}} dx.$$

Formula (3.471.9) of Gradshteyn and Ryzhik [45] includes the identity

$$\int_0^\infty x^{\varsigma_1 - 1} e^{-\frac{\varsigma_2}{x} - \varsigma_3 x} dx = 2\left(\frac{\varsigma_2}{\varsigma_3}\right)^{\frac{\varsigma_1}{2}} K_{\varsigma_1}\left(2\sqrt{\varsigma_2 \varsigma_3}\right) \tag{10}$$

for $\operatorname{Re}\varsigma_2 > 0$ and $\operatorname{Re}\varsigma_3 > 0$. Using (10), we find that

$$\varphi_{\text{SHIG}(\alpha,\beta,\delta)}(u) = \frac{K_0\left(\delta\sqrt{\alpha^2 - \beta^2 - 2iu}\right)}{K_0\left(\delta\sqrt{\alpha^2 - \beta^2}\right)}. \tag{11}$$

Similarly to (11), it is easy to find that the moments $M_{a,\text{SHIG}(\alpha,\beta,\delta)}$ for $a \in \mathbb{R}$ of the SHIG distribution are determined by the equality

$$M_{a,\text{SHIG}(\alpha,\beta,\delta)} = \text{E}(\text{SHIG}^a(\alpha,\beta,\delta)) =$$

$$= \left(\frac{\delta}{\sqrt{\alpha^2 - \beta^2}}\right)^a \frac{K_a\left(\delta\sqrt{\alpha^2 - \beta^2}\right)}{K_0\left(\delta\sqrt{\alpha^2 - \beta^2}\right)}. \tag{12}$$

Next, let us discuss the characteristic function $\varphi_{\text{SH}(\mu,\sigma,\alpha,\beta,\delta)}(u)$ of the SH distribution. Since

$$\varphi_{\text{SH}(\mu,\sigma,\alpha,\beta,\delta)}(u) = \text{E}\left(\text{E}\left(e^{iu\text{SH}}|\text{SHIG}\right)\right)$$

and

$$\text{E}\left(e^{iu\text{SH}}|\text{SHIG}\right) = e^{iu\mu}\varphi_{\text{N}\left(\beta\text{SHIG},\sigma\sqrt{\text{SHIG}}\right)} = e^{iu(\mu + \beta\text{SHIG}) - \frac{1}{2}\sigma^2 u^2 \text{SHIG}},$$

we find that

$$\varphi_{\text{SH}(\mu,\sigma,\alpha,\beta,\delta)}(u) = e^{iu\mu}\varphi_{\text{SHIG}(\alpha,\beta,\delta)}\left(\beta u + \frac{i\sigma^2 u^2}{2}\right). \tag{13}$$

Applying (8), we find that the moments $M_{n,\text{SH}(\mu,\sigma,\alpha,\beta,\delta)}$ for $n \in \mathbb{N}$ of the SH distribution can be derived as

$$M_{n,\text{SH}(\mu,\sigma,\alpha,\beta,\delta)} = \text{E}\left(\mu + \beta\text{SHIG} + \sigma\text{N}\sqrt{\text{SHIG}}\right)^n =$$

$$= \sum_{k=0}^n C_n^k \mu^{n-k} \text{E}\left(\beta\text{SHIG} + \sigma\text{N}\sqrt{\text{SHIG}}\right)^k =$$

$$= \sum_{k=0}^n C_n^k \mu^{n-k} \sum_{m=0}^k C_k^m \sigma^m \beta^{k-m} \text{E}\left(\text{SHIG}^{k-\frac{m}{2}}\right) \text{E}(\text{N}^m) =$$

$$= \sum_{k=0}^n C_n^k \mu^{n-k} \sum_{l=0}^{[k/2]} C_k^{2l} \sigma^{2l} \beta^{k-2l} M_{k-l,\text{SHIG}(\alpha,\beta,\delta)}(\max\{2l-1,1\})!!, \tag{14}$$

where $[\varsigma]$ is the integer part of $\varsigma$ and the moments of the SHIG distribution are determined by (12).

Furthermore, we have from (7)–(9) that

$$f_{\mathrm{SH}(\mu,\sigma,\alpha,\beta,\delta)}(x) = \frac{1}{\sigma\sqrt{2\pi}} \int_0^\infty y^{-\frac{1}{2}} e^{-\frac{(x-\mu-\beta y)^2}{2\sigma^2 y}} f_{\mathrm{SHIG}(\alpha,\beta,\delta)}(y) dy$$

$$= \frac{Ce^{\frac{\beta(x-\mu)}{\sigma^2}}}{\sigma\sqrt{2\pi}} \int_0^\infty y^{-\frac{3}{2}} e^{-\frac{(x-\mu)^2+\sigma^2\delta^2}{2\sigma^2 y} - \frac{\beta^2+\sigma^2(\alpha^2-\beta^2)}{2\sigma^2} y} dy. \tag{15}$$

Employing (10) to (15), we find that

$$f_{\mathrm{SH}(\mu,\sigma,\alpha,\beta,\delta)}(x) = \frac{Ce^{\frac{\beta(x-\mu)}{\sigma^2}}\sqrt{2}}{\sigma\sqrt{\pi}} \left(\frac{\beta^2+\sigma^2(\alpha^2-\beta^2)}{(x-\mu)^2+\sigma^2\delta^2}\right)^{\frac{1}{4}} \times$$

$$\times \mathrm{K}_{\frac{1}{2}}\left(\frac{\left((x-\mu)^2+\sigma^2\delta^2\right)^{\frac{1}{2}}\left(\beta^2+\sigma^2(\alpha^2-\beta^2)\right)^{\frac{1}{2}}}{\sigma^2}\right).$$

Using (3), we find that

$$f_{\mathrm{SH}(\mu,\sigma,\alpha,\beta,\delta)}(x) = \frac{C}{\sqrt{(x-\mu)^2+\sigma^2\delta^2}} e^{\frac{\beta(x-\mu)-\sqrt{\left((x-\mu)^2+\sigma^2\delta^2\right)\left(\beta^2+\sigma^2(\alpha^2-\beta^2)\right)}}{\sigma^2}}.$$

Set

$$\widehat{\delta} = \sigma\delta, \qquad\qquad \widehat{\beta} = \frac{\beta}{\sigma^2}$$

and

$$\widehat{\alpha} = \frac{\sqrt{\beta^2+\sigma^2(\alpha^2-\beta^2)}}{\sigma^2} \geq |\widehat{\beta}|.$$

Then we can rewrite $f_{\mathrm{SH}(\mu,\sigma,\alpha,\beta,\delta)}(x)$ as

$$f_{\mathrm{SH}(\mu,\sigma,\alpha,\beta,\delta)}(x) = \frac{C}{\sqrt{(x-\mu)^2+\widehat{\delta}^2}} e^{\widehat{\beta}(x-\mu)-\widehat{\alpha}\sqrt{(x-\mu)^2+\widehat{\delta}^2}} =$$

$$= f_{\mathrm{SH}\left(\mu,\widehat{\alpha},\widehat{\beta},\widehat{\delta}\right)}(x). \tag{16}$$

The identity (16) is used onwards for the computation of the characteristics of the SH distribution.

## 4. Main Results

The main results of this paper give us the formulas for the cumulative semi-hyperbolic distribution functions and the first and second lower partial moments of the semi-hyperbolic distribution. To formulate the results, we should introduce extra notations.

We set

$$\mathrm{sgn}^-(\varsigma) = \begin{cases} 1 & \text{if } \varsigma > 0, \\ -1 & \text{if } \varsigma \leq 0 \end{cases} \quad \text{and} \quad \mathrm{sgn}^+(\varsigma) = \begin{cases} 1 & \text{if } \varsigma \geq 0, \\ -1 & \text{if } \varsigma < 0. \end{cases}$$

Together with it, we employ a designation (see (16) at p.25 of Srivastava and Karlsson [51])

$$\Phi_1(\varsigma_1,\varsigma_2,\varsigma_3;\varsigma_4,\varsigma_5) \tag{17}$$

for the degenerate generalized hypergeometric function (the confluent Appell function or the Humbert series), which is the double sum

$$\sum_{m=0}^{\infty} \sum_{n=0}^{\infty} \frac{(\varsigma_1)_{m+n}(\varsigma_2)_m}{m!n!(\varsigma_3)_{m+n}} \varsigma_4^m \varsigma_5^n$$

for $|\varsigma_4| < 1$, where $(\varsigma)_l$, $l \in \mathbb{N} \cup \{0\}$, is the Pochhammer's symbol. For more information on the generalized hypergeometric functions and their integral representations, we refer to Sections 1.2 and 1.3 of Srivastava and Karlsson [51], Section 4.22 of Erdélyi et al. [52] and Srivastava et al. [53]. Hypergeometric functions on the complex plane are considered in Sadykov [54,55]. Generalized hypergeometric series of matrix variables are discussed by Cuchta et al. [56,57].

The next theorem yields an analytical expression for the cumulative semi-hyperbolic distribution function.

**Theorem 1.** *The semi-hyperbolic distribution function can be calculated for $u \in \mathbb{R}$ by the identity*

$$F_{\mathrm{SH}(\mu,\sigma,\alpha,0,\delta)}(u) = I_{\{u>\mu\}} - \frac{\mathrm{sgn}^-(u-\mu)}{2\mathrm{K}_0(\alpha\delta)} \mathcal{I}\left(\frac{u-\mu}{\sigma\delta}\right) \tag{18}$$

*with the modified Bessel function of the second kind (2), where*

$$\mathcal{I}(x) = \mathrm{K}_0(\alpha\delta) - \frac{\left(2\left(\sqrt{1+x^2}-1\right)\right)^{\frac{1}{2}}}{e^{\alpha\delta}} \times$$
$$\times \Phi_1\left(\frac{1}{2}, \frac{1}{2}, \frac{3}{2}; \frac{1-\sqrt{1+x^2}}{2}, \alpha\delta\left(1-\sqrt{1+x^2}\right)\right)$$

*for the Appell function (17).*

**Proof Sketch.** The proof of Theorem 1 can be divided into three steps.

Step 1. We show that the computation of the required probability can be cut to the calculation of the integral

$$\mathcal{I}(w) = \int_w^{\infty} \frac{e^{-\phi\left(\sqrt{1+x^2}\right)}}{\sqrt{1+x^2}} dx.$$

Step 2. For $w = 0$, the integral $\mathcal{I}(w)$ is transformed so that formula 3.383.8 of Gradshteyn and Ryzhik [45] can be applied to it.

Step 3. For $w > 0$, we convert $\mathcal{I}(w)$ to a special form so that formula 4.3.24 of Erdélyi et al. [52] can be used to it. □

The complete proof is set in Appendix A (see Proof of Theorem 1 therein).

The following example illustrates how Theorem 1 works for the simplest case $u = \mu$.

**Example 5.** *We immediately have from (18) that*

$$F_{\mathrm{SH}(\mu,\sigma,\alpha,0,\delta)}(\mu) = \frac{\mathcal{I}(0)}{2\mathrm{K}_0(\alpha\delta)} = \frac{1}{2}.$$

*Since the semi-hyperbolic distribution function is continuous, the identity*

$$F_{\mathrm{SH}(\mu,\sigma,\alpha,0,\delta)}(u) = I_{\{u\geq\mu\}} - \frac{\mathrm{sgn}^+(u-\mu)}{2\mathrm{K}_0(\alpha\delta)} \mathcal{I}\left(\frac{u-\mu}{\sigma\delta}\right) \tag{19}$$

*should be satisfied as well. Indeed, we obtain from (19) that*

$$F_{\text{SH}(\mu,\sigma,\alpha,0,\delta)}(\mu) = 1 - \frac{\mathcal{I}(0)}{2\text{K}_0(\alpha\delta)} = \frac{1}{2}$$

*again.*

The lower partial moments of the semi-hyperbolic distribution are defined by the identity

$$M_{n,\text{SH}(\mu,\sigma,\alpha,0,\delta)}(u) = \text{E}\Big(\text{SH}^n(\mu,\sigma,\alpha,0,\delta)I_{\{\text{SH}(\mu,\sigma,\alpha,0,\delta)\leq u\}}\Big).$$

The theorem below provides explicit formulas for the first and second lower partial moments of the SH distribution.

**Theorem 2.** *For $u \in \mathbb{R}$, the lower partial moments*

$$M_{1,\text{SH}(\mu,\sigma,\alpha,0,\delta)}(u) = \mu F_{\text{SH}(\mu,\sigma,\alpha,0,\delta)}(u) - \frac{1}{2\widehat{\alpha}\text{K}_0(\alpha\delta)}e^{-\widehat{\alpha}\sqrt{\widehat{\delta}^2+(u-\mu)^2}} \tag{20}$$

*and*

$$M_{2,\text{SH}(\mu,\sigma,\alpha,0,\delta)}(u) = \tag{21}$$

$$= \mu^2 F_{\text{SH}(\mu,\sigma,\alpha,0,\delta)}(u) - \frac{\mu e^{-\widehat{\alpha}\sqrt{\widehat{\delta}^2+(u-\mu)^2}}}{\widehat{\alpha}\text{K}_0(\alpha\delta)} + \frac{M_{2,\text{SH}(0,\sigma,\alpha,0,\delta)}}{2} +$$

$$+ \frac{\text{sgn}(u-\mu)}{2\widehat{\alpha}\text{K}_0(\alpha\delta)}\left(\widehat{\delta}\mathcal{J}\left(\frac{u-\mu}{\widehat{\delta}}\right) - |u-\mu|e^{-\widehat{\alpha}\sqrt{\widehat{\delta}^2+(u-\mu)^2}}\right)$$

*with $\widehat{\alpha} = \sigma^{-1}\alpha$, $\widehat{\delta} = \sigma\delta$ and*

$$\mathcal{J}(x) = \frac{\sqrt{2\widehat{x}}}{e^{\alpha\delta}}\left(\Phi_1\left(\frac{1}{2},\frac{1}{2},\frac{3}{2};-\frac{\widehat{x}}{2},-\alpha\delta\widehat{x}\right) + \frac{\widehat{x}}{3}\Phi_1\left(\frac{3}{2},\frac{1}{2},\frac{5}{2};-\frac{\widehat{x}}{2},-\alpha\delta\widehat{x}\right)\right),$$

*where $\widehat{x} = \sqrt{1+x^2} - 1$.*

**Proof Sketch.** The proof of Theorem 2 can be divided into four steps.

Step 1. The general formula for the nth lower partial moment is received.

Step 2. The value of the first lower partial moment is obtained using the result of Theorem 1.

Step 3. It is shown that the computation of the second lower partial moment can be reduced to the calculation of the integral

$$\mathcal{J}(w) = \int_0^w e^{-\phi\sqrt{1+x^2}}dx.$$

Step 4. The integral $\mathcal{J}(w)$ is transformed so that formula 4.3.24 of Erdélyi et al. [52] can be employed to it.  □

The full proof is given in Appendix A (see Proof of Theorem 2 therein).

**Example 6.** *Let $u \to \infty$ in (20). Then,*

$$\lim_{u\to\infty} M_{1,\text{SH}(\mu,\sigma,\alpha,0,\delta)}(u) = \mu \lim_{u\to\infty} F_{\text{SH}(\mu,\sigma,\alpha,0,\delta)}(u) = \mu = M_{1,\text{SH}(\mu,\sigma,\alpha,0,\delta)}.$$

*Furthermore, apparently*

$$M_{2,\mathrm{SH}(0,\sigma,\alpha,0,\delta)}(0) = \frac{1}{2}M_{2,\mathrm{SH}(0,\sigma,\alpha,0,\delta)}.$$

*Employing (21), we have the same result.*

## 5. Applications

The standards of Basel III regulations order the use of monetary risk measures to control the market risk to banks. The main risk measure is called the value-at-risk (VaR). Informally, the $p$-VaR can be defined as the maximum possible loss of the final net worth after excluding all worse outcomes whose combined probability is at most $p$. In an appropriate probabilistic model, the $p$-VaR is defined as the upper $p$-quantile of the loss distribution with the minus sign. That is, if the loss is modeled by a random variable $\varsigma$, the $p$-VaR of $\varsigma$ is

$$\mathrm{VaR}_p(\varsigma) = -u_p^+, \quad \text{where } u_p^+ = \inf\{u \in \mathbb{R} : \mathrm{P}(\varsigma \leq u) \geq p\},$$

see Definition 3.3 in Artzner et al. [58] or Definition 4.40 in Föllmer and Schied [59]. Let us note that this definition of the VaR implies small values of the parameter $p$. An alternative definition, with a large $p$, is given in McNeil et al. [47], see Definition 2.10 therein.

The expected shortfall (ES) monetary risk measure can be suggested as a substitution of the VaR since the ES takes into account the tail risk. Because of it, the ES is recommended to banks by the Basel III regulations instead of the VaR, which had been ordered by Basel II. Mathematically, the ES of level $p$ is defined for a random outcome $\varsigma$ as

$$\mathrm{ES}_p(\varsigma) = -\frac{1}{p}\left[\mathrm{E}\left(\varsigma I_{\{\varsigma \leq u_p^+\}}\right) + u_p^+\left(p - \mathrm{P}\left(\varsigma \leq u_p^+\right)\right)\right].$$

A number of research papers were set to calculate the basic risk measures in various models. The computations of the measures in semi-analytical forms by the Fourier transform technique are given in Armenti et al. [60] and Drapeu et al. [61]. Analytical formulas for some specific models are provided by Ivanov [62,63] and Rockafellar and Uryasev [64]. Monte Carlo simulations of the VAR and ES were made by Chun et al. [65] and Mafusalov and Uryasev [66]. Nonparametric methods for the estimation of these basic monetary risk measures were developed in Cai and Wang [67], Chen and Tang [68] and Scailett [69].

An advanced risk measurement can be used by banks for the aim of the management and control. Although the variance is often employed for the portfolio optimization problems (see, for example, Fontana and Schweizer [70] and Schweizer et al. [71]), it is not the appropriate choice if the underlying distribution is asymmetric. Because of it, the target semivariance risk measure is applied. The target $u$ semivariance of $\varsigma$ is defined as

$$\mathrm{TS}_u(\varsigma) = \mathrm{E}\left[(\varsigma - u)^2 I_{\{\varsigma \leq u\}}\right]. \tag{22}$$

A summary of early studies about the semivariance is given in Nawrocki [72]. Performance risk measures based on the semivariance were introduced by van der Meer and Sortino [73] and van der Meer et al. [74]. The realized semivariance was studied in Barndorff-Nielsen et al. [75]. An empirical analysis was produced by Ang et al. [76]. Computations for non-Gaussian cases were made by Jarrow and Zhao [77] and Ivanov [78].

In this section, we assume that there are $n$ assets $A_{1,t}, \ldots, A_{n,t}$ in the investment portfolio whose dynamics are determined as

$$A_{l,1} - A_{l,0} = \mu_l + \sigma_l \sqrt{\mathrm{SHIG}}N_l,$$

where $\mu_l \in \mathbb{R}$, $\sigma_l > 0$, $\mathrm{SHIG} = \mathrm{SHIG}(\alpha, 0, \delta)$ is the semi-hyperbolic inverse Gaussian variable and $N_l$, $l = 1, 2, \ldots, n$ are the standard normal random variables with correlation coefficients $\rho_{ij}$ between $N_i$ and $N_j$. It is supposed that the normal distributions are inde-

pendent with the semi-hyperbolic inverse Gaussian one. Also, we propose that a non-risk asset is included in the portfolio. Namely, $A_{0,t}$ with the dynamic

$$A_{0,1} - A_{0,0} = r, \qquad r \in \mathbb{R}.$$

As mentioned in the Introduction, Daskalaki and Katris [34] have made a detailed calibration of the GH distribution for different market indices. It is found by them that the estimations of the parameter $\lambda$ are $-0.098$ and $-0.037$ for the OMXH25 and SAX indices, respectively (see Table 12 therein). We assume that the considered portfolio consists of stocks from the indices with $\lambda \approx 0$. Such portfolios should be modeled by the semi-hyperbolic distribution.

To assess the risks of portfolio $\omega = (x_0, ..., x_n) \in \mathbb{R}^{n+1}$ at the time $t = 1$, it is enough to discuss the random variable

$$X^{\omega} = \sum_{l=0}^{n} x_l \triangle A_{l,1} = x_0 r + \sum_{l=1}^{n} x_l \mu_l + \left( \sum_{l=1}^{n} x_l \sigma_l N_l \right) \sqrt{\text{SHIG}}.$$

Set

$$\mathcal{M} = x_0 r + \sum_{l=1}^{n} x_l \mu_l \qquad \text{and} \qquad \Xi = \sqrt{\sum_{i,j=1}^{n} \rho_{ij} x_i x_j \sigma_i \sigma_j}.$$

Because of

$$\sum_{l=1}^{n} x_l \sigma_l N_l \overset{Law}{=} \Xi N,$$

where N is again the standard normal distribution, we have that

$$X^{\omega} \overset{Law}{=} \mathcal{M} + \Xi N \sqrt{\text{SHIG}} = \text{SH}(\mathcal{M}, \Xi, \alpha, 0, \delta). \tag{23}$$

To compute the $p$-VaR of $X^{\omega}$, we have to find the quantile $u_p^+ = u_p$ solving an equation

$$P(X^{\omega} \leq u_p) - p = 0$$

with one of the zero-finding algorithms, see Brent [79] for details. Once again, since $X^{\omega}$ has a continuous distribution, we have that

$$\text{ES}_p(X^{\omega}) = -\frac{1}{p} \text{E}\left( X^{\omega} I_{\left\{ X^{\omega} \leq u_p \right\}} \right).$$

It issues immediately from (22) that

$$\text{TS}_u(X^{\omega}) = \text{E}\left( (X^{\omega})^2 I_{\{X^{\omega} \leq u\}} \right) - 2u \text{E}\left( X^{\omega} I_{\{X^{\omega} \leq u\}} \right) + u^2 P(X^{\omega} \leq u).$$

Implementing (23), we obtain the following corollary of Theorems 1 and 2.

**Corollary 1.** *The value-at-risk* $\text{VaR}_p(X^{\omega})$ *is determined as the solution* $u_p$ *of the equation*

$$F_{\text{SH}(\mathcal{M}, \Xi, \alpha, 0, \delta)}(u) - p = 0$$

*with the minus sign. The expected shortfall* $\text{ES}_p(X^{\omega})$ *is calculated with respect to the formula*

$$\text{ES}_p(X^{\omega}) = -\frac{1}{p} \text{M}_{1, \text{SH}(\mathcal{M}, \Xi, \alpha, 0, \delta)}(u_p).$$

*The target u semivariance* $\mathrm{TS}_u(X^\omega)$ *is computed as*

$$\mathrm{TS}_u(X^\omega) = M_{2,\mathrm{SH}(\mathcal{M},\Xi,\alpha,0,\delta)} - 2u M_{1,\mathrm{SH}(\mathcal{M},\Xi,\alpha,0,\delta)} + u^2 F_{\mathrm{SH}(\mathcal{M},\Xi,\alpha,0,\delta)}(u).$$

## 6. Numerical Analysis

In this section, we compare the normal and semi-hyperbolic distributions. The idea is to adjust the parameters of the SH distribution so that the second moments of the distributions match each other.

It is easy to see from (12) and (14) that

$$M_{2,\mathrm{SH}(0,\sigma,\alpha,0,\delta)} = \sigma^2 M_{1,\mathrm{SHIG}(\alpha,0,\delta)} = \frac{\delta \sigma^2 \mathrm{K}_1(\alpha\delta)}{\alpha \mathrm{K}_0(\alpha\delta)}.$$

The identity

$$M_{2,\mathrm{SH}(0,\sigma,\alpha,0,\delta)} = M_{2,\mathrm{N}(0,\sigma)}$$

holds if

$$\frac{\delta \mathrm{K}_1(\alpha\delta)}{\alpha \mathrm{K}_0(\alpha\delta)} = 1.$$

Set $\alpha = 1$. Then, the sought $\delta^*$ should solve the equation

$$\delta = \frac{\mathrm{K}_0(\delta)}{\mathrm{K}_1(\delta)}.$$

The tables of the values of the modified Bessel function of the second kind give us the solution

$$\delta^* \approx 0.595.$$

Therefore, we compare the distributions

$$\mathrm{N}(0,\sigma) \qquad \text{and} \qquad \mathrm{SH}(0,\sigma,1,0,0.595). \tag{24}$$

Set $\sigma = 1$. Then, the probability density functions of these normal and semi-hyperbolic distributions are

$$f_{\mathrm{N}(0,1)} = \frac{1}{\sqrt{2\pi}} e^{-\frac{x^2}{2}} \qquad \text{and} \qquad f_{\mathrm{SH}(0,1,1,0,\delta^*)} = \frac{e^{-\sqrt{x^2+(\delta^*)^2}}}{2\mathrm{K}_0(\delta^*)\sqrt{x^2+(\delta^*)^2}},$$

respectively. The density functions are plotted at Figure 1.

Furthermore, it follows from (18) that

$$F_{\mathrm{SH}(0,1,1,0,\delta^*)}(u) = I_{\{u>0\}} - \frac{\mathrm{sgn}^-(u)}{2\mathrm{K}_0(\delta^*)}\mathcal{I}\left(\frac{u}{\delta^*}\right)$$

with

$$\mathcal{I}\left(\frac{u}{\delta^*}\right) = \mathrm{K}_0(\delta^*) - \frac{\left(2\left(\sqrt{1+\left(\frac{u}{\delta^*}\right)^2}-1\right)\right)^{\frac{1}{2}}}{e^{\delta^*}} \times$$

$$\times \Phi\left(\frac{1}{2}, \frac{1}{2}, \frac{3}{2}; \frac{1-\sqrt{1+\left(\frac{u}{\delta^*}\right)^2}}{2}, \delta^*\left(1-\sqrt{1+\left(\frac{u}{\delta^*}\right)^2}\right)\right).$$

Formula (4.3.24) of Erdélyi et al. [52] includes the identity

$$\Phi(\varsigma_1, \varsigma_2, \varsigma_3; \varsigma_4, \varsigma_5) = \tag{25}$$

$$= \frac{1}{B(\varsigma_1, \varsigma_3 - \varsigma_1)} \int_0^1 x^{\varsigma_1 - 1}(1-x)^{\varsigma_3 - \varsigma_1 - 1}(1 - \varsigma_4 x)^{-\varsigma_2} e^{\varsigma_5 x} dx$$

for $\varsigma_1 > 0$, $\varsigma_3 > u_1$ and $\varsigma_4 < 1$. We have from (25) that

$$\Phi\left(\frac{1}{2}, \frac{1}{2}, \frac{3}{2}; A, B\right) = \frac{1}{2}J(A, B),$$

where

$$J(A, B) = \int_0^1 x^{-\frac{1}{2}}(1 - Ax)^{-\frac{1}{2}} e^{Bx} dx.$$

Hence, we find that for $u \leq 0$

$$F_{\text{SH}(0,1,1,0,\delta^*)}(u) = \frac{1}{2} - \frac{J\left(-\frac{u^*}{2}, -\delta^* u^*\right)\sqrt{u^*}}{2e^{\delta^*} K_0(\delta^*)\sqrt{2}},$$

where

$$u^* = \sqrt{1 + \left(\frac{u}{\delta^*}\right)^2} - 1.$$

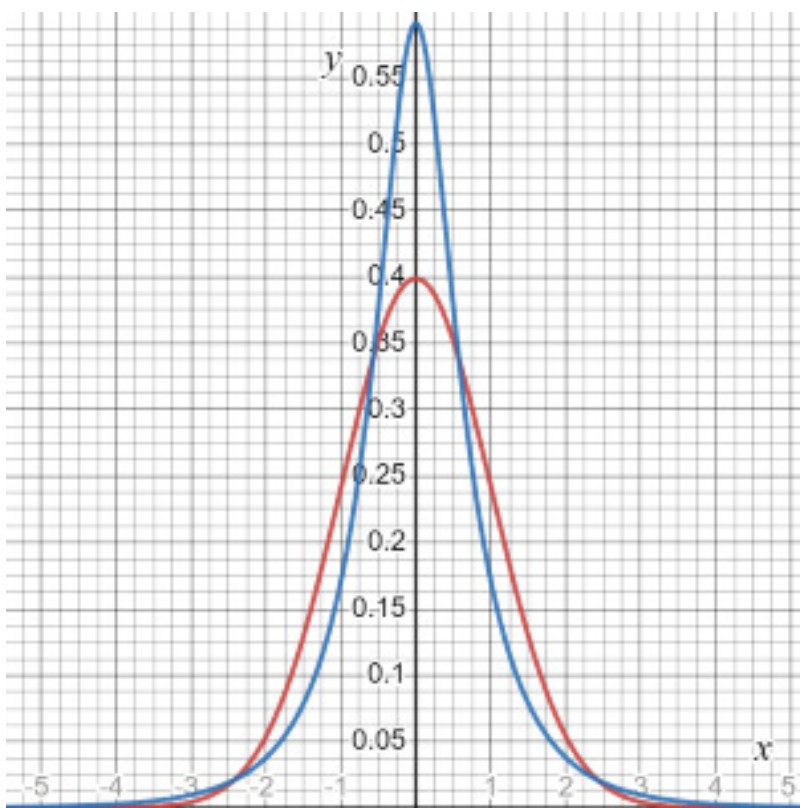

**Figure 1.** The normal (red) and semi-hyperbolic (blue) probability density functions.

Based on the calculations above, Table 1 provides the comparison of the symmetric normal and semi-hyperbolic distributions (24) with the same mean and variance for variant arguments.

**Table 1.** The comparison of the cumulative distribution functions.

| $F \backslash u$ | $-3$ | $-2$ | $-1$ | $-0.5$ | $-0.2$ | $0$ |
|---|---|---|---|---|---|---|
| N | 0.00135 | 0.02275 | 0.15866 | 0.30854 | 0.42074 | 0.5 |
| SH | 0.00783 | 0.02841 | 0.11551 | 0.2535 | 0.38514 | 0.5 |

## 7. Discussion

The obtained results show that the analytical expressions based on values of generalized hypergeometric functions (Sections 1.2 and 1.3 of Srivastava and Karlsson [51]) are approachable for the GH models besides the NIG one (Ivanov and Temnov [10]). The derived formulas use the modified Bessel function of the second kind, the Appell function, and can be computed under one second with the modern software. As in the VG model (Madan et al. [30], Ivanov [63]), the explicit solutions are obtained for the cumulative distribution function, the first and second lower partial moments of the SH distribution, and the received formulas are applied to the problem of computation of the VAR, the ES, and the semivariance in the related investment portfolio model with dependent asset returns.

Numerical computations, which are provided for the symmetric normal and SH distributions with the same mean and variance have discovered that the cumulative distribution functions significantly differ at the tails and mid part. The SH distribution has the tails for the arguments $-3$ and $3$, which are six times larger than the idem tails of the normal one. And at the same time, the mid part (between $-0.5$ and $0.5$) of the SH distribution relates to the similar part of the normal one as 5 to 4. Therefore, we deduce that the SH model distinguishes better phenomena with a larger number of extreme events and events close to the mean value.

A future study should yield the computation of the nth lower partial moments of SH distribution together with the calculation of the prices of option contracts in the SH model since such results are available in the VG model, see Madan et al. [30] and Ivanov [78]. The SH Lévy process with drift switching (Ivanov [63]) can be studied. Together with it, the next investigations should relate to the obtaining of analytical solutions for the GH hyperbolic distributions with integer $\lambda$ including the hyperbolic one of Eberlein and Keller [13]. Although the standard stochastic simulation methods (Chapter XII of Asmussen and Glynn [80]) can be applied to the semi-hyperbolic distribution, it is necessary to notice that their speed is often insufficient. Because of this, an interesting problem is to develop fast quasi-Monte Carlo procedures for the SH distribution similarly to the methods that are developed for the variance-gamma distribution (Avramidis et al. [81], Avaramidis and L'Ecuyer [82] and Chapter 6.2 of Lemieux [83]).

## 8. Conclusions

Taking into account the review of the related literature, the results of this paper, including the applications and the numerical analysis, and the discussion of the results, we can make the following conclusions.

- The review of the literature about the class of GH distribution confirms the necessity of the development of the mathematical methods of its analysis.
- The subclass of the family of GH distributions, the semi-hyperbolic distributions, is analytically tractable similarly to the VG distributions.
- The obtained formulas depend on the values of degenerate generalized hypergeometric functions and can be computed very fast.
- The numerical analysis shows that the SH distribution discerns better than the normal data with heavy tails and a central part.
- Keeping in mind the amount of work, we look forward to the future studies of the whole class of GH distribution.

**Funding:** This research received no external funding.

**Institutional Review Board Statement:** Not applicable.

**Informed Consent Statement:** Not applicable.

**Data Availability Statement:** Not applicable.

**Acknowledgments:** I would like to thank the anonymous referees whose remarks have significantly improved the paper.

**Conflicts of Interest:** The author declares no conflict of interest.

## Abbreviations

The following abbreviations are used in this manuscript:

| | |
|---|---|
| GH | Generalized Hyperbolic |
| SH | Semi-Hyperbolic |
| GIG | Generalized Inverse Gaussian |
| NIG | Normal-Inverse Gaussian |
| SHIG | Semi-Hyperbolic Inverse Gaussian |
| VaR | Value-at-Risk |
| ES | Expected Shortfall |

## Appendix A

**Proof of Theorem 1.** Using (16) with $C$ defined by (9), it is easy to notice that

$$F_{\mathrm{SH}(\mu,\sigma,\alpha,0,\delta)}(u) = \int_{-\infty}^{u} f_{\mathrm{SH}(\mu,\sigma,\alpha,0,\delta)}(x)dx =$$

$$= \int_{-\infty}^{u} f_{\mathrm{SH}(\mu,\widehat{\alpha},0,\widehat{\delta})}(x)dx = C \int_{-\infty}^{u} \frac{e^{-\widehat{\alpha}\sqrt{(x-\mu)^2+\widehat{\delta}^2}}}{\sqrt{(x-\mu)^2+\widehat{\delta}^2}}dx =$$

$$= C \int_{-\infty}^{u-\mu} \frac{e^{-\widehat{\alpha}\sqrt{x^2+\widehat{\delta}^2}}}{\sqrt{x^2+\widehat{\delta}^2}}dx = CJ(\mu-u), \tag{A1}$$

where

$$J(v) = \int_{v}^{\infty} \frac{e^{-\widehat{\alpha}\sqrt{x^2+\widehat{\delta}^2}}}{\sqrt{x^2+\widehat{\delta}^2}}dx.$$

Next, we compute $J(v)$ for variant values of $v$.

Case 1. $v \geq 0$. Set

$$w = \frac{v}{\widehat{\delta}} \qquad \text{and} \qquad \phi = \widehat{\alpha}\widehat{\delta}.$$

Then,

$$J(v) = \mathcal{I}(w), \tag{A2}$$

where

$$\mathcal{I}(w) = \int_{w}^{\infty} \frac{e^{-\phi\left(\sqrt{1+x^2}\right)}}{\sqrt{1+x^2}}dx$$

with $w \geq 0$. Therefore,

$$F_{\mathrm{SH}(\mu,\sigma,\alpha,0,\delta)}(u) = C\mathcal{I}\left(\frac{\mu-u}{\widehat{\delta}}\right) \tag{A3}$$

when $u \leq \mu$.

Put $y = \sqrt{1 + x^2}$. Thereat

$$x = \sqrt{y^2 - 1}, \qquad\qquad dx = \frac{y}{\sqrt{y^2 - 1}} dy$$

and therefore

$$\mathcal{I}(w) = \int_{\sqrt{1+w^2}}^{\infty} \frac{e^{-\phi y}}{\sqrt{y^2 - 1}} dy = e^{-\phi} \int_{\sqrt{1+w^2}-1}^{\infty} e^{-\phi y} y^{-\frac{1}{2}} (y+2)^{-\frac{1}{2}} dy. \tag{A4}$$

Case 1.1. $w = 0$. Then,

$$\mathcal{I}(w) = \mathcal{I}(0) = e^{-\phi} \int_{0}^{\infty} e^{-\phi y} y^{-\frac{1}{2}} (y+2)^{-\frac{1}{2}} dy. \tag{A5}$$

Formula (3.383.8) of Gradshteyn and Ryzhik [45] includes the identity

$$\int_{0}^{\infty} x^{\varsigma_1 - 1} (x + \varsigma_2)^{\varsigma_1 - 1} e^{-\varsigma_3 x} dx = \tag{A6}$$

$$= \frac{1}{\sqrt{\pi}} \left( \frac{\varsigma_2}{\varsigma_3} \right)^{\varsigma_1 - \frac{1}{2}} e^{\frac{\varsigma_2 \varsigma_3}{2}} \Gamma(\varsigma_1) \mathrm{K}_{\frac{1}{2} - \varsigma_1} \left( \frac{\varsigma_2 \varsigma_3}{2} \right)$$

for $\varsigma_1 > 0$, $\varsigma_2 > 0$, $\varsigma_3 > 0$. Applying (A6) to (A5), we find that

$$\mathcal{I}(0) = \frac{1}{\sqrt{\pi}} \Gamma\left( \frac{1}{2} \right) \mathrm{K}_0(\phi) = \mathrm{K}_0(\phi). \tag{A7}$$

Case 1.2. $w > 0$. Let

$$z = \frac{y}{\sqrt{1 + w^2} - 1}.$$

Then it follows from (A4) that

$$\mathcal{I}(w) = e^{-\phi} \int_{1}^{\infty} z^{-\frac{1}{2}} \left( z + \frac{2}{\sqrt{1+w^2}-1} \right)^{-\frac{1}{2}} e^{-\phi \left( \sqrt{1+w^2}-1 \right) z} dz =$$
$$= \mathcal{I}_1(w) - \mathcal{I}_2(w), \tag{A8}$$

where

$$\mathcal{I}_1(w) = e^{-\phi} \int_{0}^{\infty} z^{-\frac{1}{2}} \left( z + \frac{2}{\sqrt{1+w^2}-1} \right)^{-\frac{1}{2}} e^{-\phi \left( \sqrt{1+w^2}-1 \right) z} dz$$

and

$$\mathcal{I}_2(w) = e^{-\phi} \int_{0}^{1} z^{-\frac{1}{2}} \left( z + \frac{2}{\sqrt{1+w^2}-1} \right)^{-\frac{1}{2}} e^{-\phi \left( \sqrt{1+w^2}-1 \right) z} dz.$$

According to (A6), we find that

$$\mathcal{I}_1(w) = \mathrm{K}_0(\phi). \tag{A9}$$

Next, we find that

$$\mathcal{I}_2(w) = \tag{A10}$$

$$= e^{-\phi} \left( \frac{\sqrt{1+w^2}-1}{2} \right)^{\frac{1}{2}} \int_{0}^{1} z^{-\frac{1}{2}} \left( 1 + \frac{z \left( \sqrt{1+w^2}-1 \right)}{2} \right)^{-\frac{1}{2}} e^{-\phi \left( \sqrt{1+w^2}-1 \right) z} dz.$$

Formula (4.3.24) of Erdélyi et al. [52] comprises the identity

$$\int_0^1 x^{\varsigma_1-1}(1-x)^{\varsigma_2-1}(1-\varsigma_4 x)^{-\varsigma_3}e^{\varsigma_5 x}dx = \tag{A11}$$

$$= \mathrm{B}(\varsigma_1,\varsigma_2)\Phi_1(\varsigma_1,\varsigma_3,\varsigma_1+\varsigma_2;\varsigma_4,\varsigma_5)$$

for $\varsigma_1 > 0$, $\varsigma_2 > 0$ and $\varsigma_4 < 1$. Employing (A11) to (A10), we find that

$$\mathcal{I}_2(w) = e^{-\phi}\left(\frac{\sqrt{1+w^2}-1}{2}\right)^{\frac{1}{2}}\mathrm{B}\left(\frac{1}{2},1\right)\times$$

$$\times\Phi\left(\frac{1}{2},\frac{1}{2},\frac{3}{2};\frac{1-\sqrt{1+w^2}}{2},\phi\left(1-\sqrt{1+w^2}\right)\right). \tag{A12}$$

Case 2. $v < 0$. Then,

$$J(v) = \mathrm{C}^{-1} - \int_{-\infty}^v \frac{e^{-\widehat{\alpha}\sqrt{x^2+\widehat{\delta}^2}}}{\sqrt{x^2+\widehat{\delta}^2}}dx = \mathrm{C}^{-1} - J(-v)$$

and we can use Case 1 herein. Then,

$$F_{\mathrm{SH}(\mu,\sigma,\alpha,0,\delta)}(u) = 1 - \mathrm{C}\mathcal{I}\left(\frac{u-\mu}{\widehat{\delta}}\right) \tag{A13}$$

if $u > \mu$.

Finally, we summarize (A3) and (A13) employing (A7)–(A9) and (A12). We elicit that

$$F_{\mathrm{SH}(\mu,\sigma,\alpha,0,\delta)}(u) = \mathrm{C}\mathcal{I}\left(\frac{\mu-u}{\widehat{\delta}}\right)I_{\{u\le\mu\}} + \left(1 - \mathrm{C}\mathcal{I}\left(\frac{u-\mu}{\widehat{\delta}}\right)\right)I_{\{u>\mu\}}, \tag{A14}$$

where

$$\mathcal{I}(x) = \mathrm{K}_0(\phi)I_{\{x=0\}} + \left(\mathrm{K}_0(\phi) - e^{-\phi}\left(\frac{\sqrt{1+x^2}-1}{2}\right)^{\frac{1}{2}}\times\right.$$

$$\left.\times\mathrm{B}\left(\frac{1}{2},1\right)\Phi\left(\frac{1}{2},\frac{1}{2},\frac{3}{2};\frac{1-\sqrt{1+x^2}}{2},\phi\left(1-\sqrt{1+x^2}\right)\right)\right)I_{\{x>0\}}$$

for $x \ge 0$. We have (18) immediately from (A14).  $\square$

**Proof of Theorem 2.** At first, let us discuss the nth lower partial moment of the semi-hyperbolic distribution for $n \in \mathbb{N}\cup\{0\}$. We have from (16) that

$$M_{n,\mathrm{SH}(\mu,\sigma,\alpha,0,\delta)}(u) = \int_{-\infty}^u x^n f_{\mathrm{SH}(\mu,\widehat{\alpha},0,\widehat{\delta})}(x)dx =$$

$$= \mathrm{C}\int_{-\infty}^u \frac{x^n e^{-\widehat{\alpha}\sqrt{\widehat{\delta}^2+(x-\mu)^2}}}{\sqrt{\widehat{\delta}^2+(x-\mu)^2}}dx = \mathrm{C}\int_{-\infty}^u \frac{(x-\mu+\mu)^n e^{-\widehat{\alpha}\sqrt{\widehat{\delta}^2+(x-\mu)^2}}}{\sqrt{\widehat{\delta}^2+(x-\mu)^2}}dx =$$

$$= \mathrm{C}\sum_{k=0}^n \mathrm{C}_n^k\mu^k H_{n-k}(u-\mu), \tag{A15}$$

where

$$H_m(v) = \int_{-\infty}^v \frac{x^m e^{-\widehat{\alpha}\sqrt{\widehat{\delta}^2+x^2}}}{\sqrt{\widehat{\delta}^2+x^2}}dx,$$

$m \in \mathbb{N}\cup\{0\}$.

Next, we pass to the computation of the first lower partial moment. We find that

$$H_0(v) = C^{-1} F_{\mathrm{SH}(0,\sigma,\alpha,0,\delta)}(v). \tag{A16}$$

Since

$$\left(e^{-\widehat{\alpha}\sqrt{\widehat{\delta}^2+x^2}}\right)'_x = -\frac{\widehat{\alpha} x e^{-\widehat{\alpha}\sqrt{\widehat{\delta}^2+x^2}}}{\sqrt{\widehat{\delta}^2+x^2}},$$

it is easy to notice that

$$H_1(v) = -\frac{e^{-\widehat{\alpha}\sqrt{\widehat{\delta}^2+v^2}}}{\widehat{\alpha}}. \tag{A17}$$

Therefore, we find that

$$M_{1,\mathrm{SH}(\mu,\sigma,\alpha,0,\delta)}(u) = \mu F_{\mathrm{SH}(0,\sigma,\alpha,0,\delta)}(u-\mu) - \frac{C}{\widehat{\alpha}} e^{-\widehat{\alpha}\sqrt{\widehat{\delta}^2+(u-\mu)^2}} =$$

$$= \mu F_{\mathrm{SH}(\mu,\sigma,\alpha,0,\delta)}(u) - \frac{C}{\widehat{\alpha}} e^{-\widehat{\alpha}\sqrt{\widehat{\delta}^2+(u-\mu)^2}} \tag{A18}$$

and obtain (20).

Furthermore, we calculate the second lower partial moment of the semi-hyperbolic distribution. We find that

$$H_2(v) = \frac{C^{-1}}{2} M_{2,\mathrm{SH}(0,\sigma,\alpha,0,\delta)} + \mathrm{sgn}(v)\widehat{H}_2(v) \tag{A19}$$

with

$$\widehat{H}_2(v) = \int_0^{|v|} \frac{x^2 e^{-\widehat{\alpha}\sqrt{\widehat{\delta}^2+x^2}}}{\sqrt{\widehat{\delta}^2+x^2}} dx.$$

Integrating by parts, we find that

$$\widehat{H}_2(v) = -\frac{1}{\widehat{\alpha}} \int_0^{|v|} x d e^{-\widehat{\alpha}\sqrt{\widehat{\delta}^2+x^2}} = \frac{1}{\widehat{\alpha}} \int_0^{|v|} e^{-\widehat{\alpha}\sqrt{\widehat{\delta}^2+x^2}} dx -$$

$$- \frac{|v| e^{-\widehat{\alpha}\sqrt{\widehat{\delta}^2+v^2}}}{\widehat{\alpha}} = \frac{\widehat{\delta}}{\widehat{\alpha}} \mathcal{J}\left(\frac{|v|}{\widehat{\delta}}\right) - \frac{|v| e^{-\widehat{\alpha}\sqrt{\widehat{\delta}^2+v^2}}}{\widehat{\alpha}} \tag{A20}$$

with

$$\mathcal{J}(w) = \int_0^w e^{-\phi\sqrt{1+x^2}} dx,$$

where $w \geq 0$.

Let $y = \sqrt{1+x^2}$. Then,

$$x = \sqrt{y^2-1}, \qquad\qquad dx = \frac{y dy}{\sqrt{y^2-1}}$$

and hence

$$\mathcal{J}(w) = \int_1^{\sqrt{1+w^2}} y(y^2-1)^{-\frac{1}{2}} e^{-\phi y} dy.$$

Set $z = y - 1$. Then,

$$\mathcal{J}(w) = e^{-\phi} \int_0^{\sqrt{1+w^2}-1} (z+1)(z+2)^{-\frac{1}{2}} z^{-\frac{1}{2}} e^{-\phi z} dz = \mathcal{J}_1(w) + \mathcal{J}_2(w), \tag{A21}$$

where

$$\mathcal{J}_1(w) = e^{-\phi} \int_0^{\widehat{w}} (z+2)^{-\frac{1}{2}} z^{-\frac{1}{2}} e^{-\phi z} dz$$

and

$$\mathcal{J}_2(w) = e^{-\phi} \int_0^{\widehat{w}} (z+2)^{-\frac{1}{2}} z^{\frac{1}{2}} e^{-\phi z} dz$$

with

$$\widehat{w} = \sqrt{1 + w^2} - 1. \tag{A22}$$

Put $h = z\widehat{w}^{-1}$ for $\widehat{w} > 0$. Thereat

$$\mathcal{J}_1(w) = e^{-\phi} \left(\frac{\widehat{w}}{2}\right)^{\frac{1}{2}} \int_0^1 h^{-\frac{1}{2}} \left(1 + \frac{\widehat{w}}{2}h\right)^{-\frac{1}{2}} e^{-\phi\widehat{w}h} dh \tag{A23}$$

and

$$\mathcal{J}_2(w) = e^{-\phi}\widehat{w} \left(\frac{\widehat{w}}{2}\right)^{\frac{1}{2}} \int_0^1 h^{\frac{1}{2}} \left(1 + \frac{\widehat{w}}{2}h\right)^{-\frac{1}{2}} e^{-\phi\widehat{w}h} dh. \tag{A24}$$

Employing (A11) to (A23) and (A24), we deduce that

$$\mathcal{J}_1(w) = e^{-\phi} \left(\frac{\widehat{w}}{2}\right)^{\frac{1}{2}} B\left(\frac{1}{2},1\right) \Phi_1\left(\frac{1}{2},\frac{1}{2},\frac{3}{2};-\frac{\widehat{w}}{2},-\phi\widehat{w}\right) \tag{A25}$$

and

$$\mathcal{J}_2(w) = e^{-\phi}\widehat{w} \left(\frac{\widehat{w}}{2}\right)^{\frac{1}{2}} B\left(\frac{3}{2},1\right) \Phi_1\left(\frac{3}{2},\frac{1}{2},\frac{5}{2};-\frac{\widehat{w}}{2},-\phi\widehat{w}\right). \tag{A26}$$

Hence, we have from (A21), (A25) and (A26) that

$$\mathcal{J}(w) = \tag{A27}$$

$$= \frac{\sqrt{2\widehat{w}}}{e^{\phi}} \left(\Phi_1\left(\frac{1}{2},\frac{1}{2},\frac{3}{2};-\frac{\widehat{w}}{2},-\phi\widehat{w}\right) + \frac{\widehat{w}}{3}\Phi_1\left(\frac{3}{2},\frac{1}{2},\frac{5}{2};-\frac{\widehat{w}}{2},-\phi\widehat{w}\right)\right).$$

Combining together (A19), (A20) and (A27), we find that

$$H_2(v) = \frac{C^{-1}}{2} M_{2,\mathrm{SH}(0,\sigma,\alpha,0,\delta)} + \tag{A28}$$

$$+ \frac{\mathrm{sgn}(v)}{\widehat{\alpha}} \left(\widehat{\delta}\mathcal{J}\left(\frac{|v|}{\widehat{\delta}}\right) - |v|e^{-\widehat{\alpha}\sqrt{\widehat{\delta}^2+v^2}}\right) I_{\{v\neq 0\}}.$$

Now, we conclude from (A15)–(A17) and (A28) that

$$M_{2,\mathrm{SH}(\mu,\sigma,\alpha,0,\delta)}(u) = C\left(H_2(u-\mu) + 2\mu H_1(u-\mu) + \mu^2 H_0(u-\mu)\right) =$$

$$= \mu^2 F_{\mathrm{SH}(0,\sigma,\alpha,0,\delta)}(u-\mu) - \frac{2\mu C e^{-\widehat{\alpha}\sqrt{\widehat{\delta}^2+(u-\mu)^2}}}{\widehat{\alpha}} + \frac{M_{2,\mathrm{SH}(0,\sigma,\alpha,0,\delta)}}{2} +$$

$$+ \frac{\mathrm{sgn}(u-\mu)C}{\widehat{\alpha}} \left(\widehat{\delta}\mathcal{J}\left(\frac{|u-\mu|}{\widehat{\delta}}\right) - |u-\mu|e^{-\widehat{\alpha}\sqrt{\widehat{\delta}^2+(u-\mu)^2}}\right) I_{\{u\neq\mu\}}$$

and obtain (21). □

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
