# Peer review of "The Semi-Hyperbolic Distribution and Its Applications"

_stats, doi:10.3390/stats6040071_

Round 1

Reviewer 1 Report

The author proposes yet another special case of the well-known generalised hyperbolic distributions, that he denotes the semi-hyperbolic distribution, which, to my knowledge has not been discussed earlier. However, I find no convincing arguments for why this particular special case is useful. There is nothing on how to estimate its parameters or how to simulate from it. There is no discussion of what characteristics this distribution has that  separates it from the other special cases, and that would make it useful. The only comparison that is made, is to the normal distribution, which is not particularly informative as the mixing obviously makes it different from the normal distribution; that is the whole point of the generalised hypoerbolic distributions to begin with. Further, the example with the financial portfolio seems artificial. One has to impose rather strong conditions on the distributions of the assets, following from a common mixing variable, to obtain a distribution for the portfolio, for which one may do computations by hand. Why would one do that in these days, when accurate results may be obtained quickly by a combination of simulation and Monte Carlo methods, without having to make unrealistic assumptions. All in all, I find that the paper has little interest and relevance, and I therefore recommend to reject it.

The language should be improved, but is overall not too bad.

Author Response

Dear the reviewer, I have uploaded the response file.

Reviewer 2 Report

The article starts with a clear and well-defined objective: to study semi-hyperbolic distributions and analyze their properties, particularly in the context of financial markets. This clarity in purpose is crucial for guiding the reader through the research. Further, it makes a valuable contribution by deriving analytical expressions for the cumulative distribution function and the first and second lower partial moments of semi-hyperbolic distributions. These expressions are essential tools for understanding and working with these distributions.

I would like to recommend the paper for publication after the following observations are taken care of:

1. The references are not cited in the order.

2. While the paper's objective is clear, the presentation of the introduction should be made more clear so that the improved clarity and structure, would make it easier for readers to follow the research flow.

3.  The paper mentions that formulas are derived, but it could be enhanced by providing step-by-step explanations or derivations for these formulas. This would aid readers in understanding the mathematical processes involved.

4.  Consider practical examples or case studies from the financial industry. Real-world scenarios can help readers connect theory to practice.

Author Response

(The authors gave the same response as above.)

Reviewer 3 Report

This research studies a subclass of generalized hyperbolic distributions which is called the semi-hyperbolic distributions. Analytical expressions for the cumulative distribution function, the first and second lower partial moments of them are obtained. Using the received formulas, the author computes the value-at-risk, the expected shortfall and the semivariance in the semi-hyperbolic mode of financial market.

The introduction is well written and contains a wide range of useful references comcerning with the topic of the article including economical grounds. Then the author outlines the structure of the whole paper. 

Section 2 is extremely useful since it describes methods exploiting in the article and has four examples of known distribution:  Inverse Gaussian distribution,  Hyperbolic inverse Gaussian distribution, . Gamma distribution and  Inverse gamma distribution.

Section 3 comprises two subsections: preliminary formulas and main results. I may recommend to divide this section into two independent subsetion unless there is an explanation why the author needs nested subsections. 

In line 162 the author mentioned Appell function without any references to a kind of them (one of four Appell functions). 

Section 3.2 has two theorems which are called the main results of the paper. Unfortunately, the proofs of the theorems are omitted. Can you provide a reader with some references or proofs? It is not obvious. 

Finally, I would like to recommend the author some recent papers by Timur Sadykov and Tom Cuchta, connected with hypergeometric functions. It might be a good source of additional references. 

Author Response

(The authors gave the same response as above.)
